

# Molecular assessment of the fecal microbiota in healthy cats and dogs before and during supplementation with fructo-oligosaccharides (FOS) and inulin using high-throughput 454-pyrosequencing

Jose F. Garcia-Mazcorro[1,2], Jose R. Barcenas-Walls[3], Jan S. Suchodolski[1] and Jörg M. Steiner[1]

[1] Gastrointestinal Laboratory, Department of Small Animal Clinical Sciences, College of Veterinary Medicine & Biomedical Sciences, Texas A&M University, College Station, TX, United States

[2] Faculty of Veterinary Medicine, Universidad Autónoma de Nuevo León, General Escobedo, Nuevo Leon, Mexico

[3] Center for Research and Development in Health Sciences (CIDICS), Genomics Unit, Universidad Autónoma de Nuevo León, Monterrey, Nuevo Leon, Mexico

Corresponding author
Jose F. Garcia-Mazcorro, josegarcia_mex@hotmail.com

## ABSTRACT

Prebiotics are selectively fermentable dietary compounds that result in changes in the composition and/or activity of the intestinal microbiota, thus conferring benefits upon host health. In veterinary medicine, commercially available products containing prebiotics have not been well studied with regard to the changes they trigger on the composition of the gut microbiota. This study evaluated the effect of a commercially available nutraceutical containing fructo-oligosaccharides (FOS) and inulin on the fecal microbiota of healthy cats and dogs when administered for 16 days. Fecal samples were collected at two time points before and at two time points during prebiotic administration. Total genomic DNA was obtained from fecal samples and 454-pyrosequencing was used for 16S rRNA gene bacterial profiling. The linear discriminant analysis (LDA) effect size (LEfSe) method was used for detecting bacterial taxa that may respond (i.e., increase or decrease in its relative abundance) to prebiotic administration. Prebiotic administration was associated with a good acceptance and no side effects (e.g., diarrhea) were reported by the owners. A low dose of prebiotics (50 mL total regardless of body weight with the end product containing 0.45% of prebiotics) revealed a lower abundance of Gammaproteobacteria and a higher abundance of Veillonellaceae during prebiotic administration in cats, while Staphylococcaceae showed a higher abundance during prebiotic administration in dogs. These differences were not sufficient to separate bacterial communities as shown by analysis of weighted UniFrac distance metrics. A predictive approach of the fecal bacterial metagenome using Phylogenetic Investigation of Communities by Reconstruction of Unobserved States (PICRUSt) also did not reveal differences between the period before and during prebiotic administration. A second trial using a higher dose of prebiotics (3.2 mL/kg body weight with the end product containing 3.1% of prebiotics) was tested in dogs and revealed a lower abundance of *Dorea* (family Clostridiaceae) and a higher abundance of *Megamonas* and other

(unknown) members of Veillonellaceae during prebiotic administration. Again, these changes were not sufficient to separate bacterial communities or predicted metabolic profiles according to treatment. A closer analysis of bacterial communities at all time-points revealed highly individualized patterns of variation. This study shows a high interindividual variation of fecal bacterial communities from pet cats and dogs, that these communities are relatively stable over time, and that some of this variation can be attributable to prebiotic administration, a phenomenon that may be affected by the amount of the prebiotic administered in the formulation. This study also provides insights into the response of gut bacterial communities in pet cats and dogs during administration of commercially available products containing prebiotics. More studies are needed to explore potentially beneficial effects on host health beyond changes in bacterial communities.

# INTRODUCTION

The digestive tract of cats and dogs is inhabited by millions of microorganisms (especially bacteria) that exert a positive and vital effect on host health (*Suchodolski, 2011*). A large number of articles are steadily being published showing the extent (e.g., in microbial composition) and consequences (e.g., relationship of specific microbes with persistence of clinical signs) of this symbiosis in health and during a variety of disease states and conditions such as obesity, gastrointestinal inflammation, and diarrhea (*Deusch et al., 2015*; *Guard et al., 2015*; *Hand et al., 2013*; *Handl et al., 2013*; *Junginger et al., 2014*; *Kieler et al., 2016*; *Minamoto et al., 2014*; *Minamoto et al., 2015*; *Song et al., 2013*; *Suchodolski et al., 2015*). These studies are supported by meta'omic analytic techniques (*Morgan & Huttenhower, 2014*) and powerful freely-available computational resources to analyze the generated data (*Navas-Molina et al., 2013*).

Humans and other mammals, such as cats and dogs, do not have all the necessary enzymes in their small intestinal tract capable of degrading several types of plant fibers (*Flint et al., 2012*). Upon consumption and after traveling throughout the small intestine, some types of these non-digestible fibers (e.g., fructo-oligosaccharides) are fermented by the bacterial microbiota in the colon thus exerting a positive effect on the abundance of beneficial bacterial groups (e.g., *Lactobacillus* and *Bifidobacterium*), intestinal motility, epithelial cellular integrity, and microbial biochemical networks (*Scott et al., 2015*). Interestingly, prebiotics appear to also influence distant sites such as bones and skin, apparently through an increase of beneficial bacteria in the gut, and derived fermentation products from this increase reaching target cells (*Collins & Reid, 2016*). Several research studies have shown beneficial effects associated with the consumption of fiber on gut microbiota and overall health (e.g., improvement of gut barrier integrity) in humans and other vertebrates (*Montalban-Arques et al., 2015*).

Prebiotics are non-digestible carbohydrates such as fructo-oligosaccharides (FOS), galacto-oligosaccharides (GOS) and inulin that are currently added to several commercial foods for cats and dogs. Studies have shown an effect of these ingredients on fecal microbial composition, nutrient digestibility, and short-chain fatty acid concentrations, particularly in dogs (*Patra, 2011*; *Schmitz & Suchodolski, 2016*; *De Godoy, Kerr & Fahey, 2013*). Domestic cats are obligate carnivores but several studies support the hypothesis that microbial fermentation inside the distal gut is significant and beneficial to the host (*Rochus, Janssens & Hesta, 2014*). However, most of the published studies have researched the effect of natural prebiotics (with and without processing, e.g., potato fiber, see *Panasevich et al., 2015*) as opposed to commercial preparations containing these ingredients. This generates an important gap in the prebiotic literature because commercial prebiotic preparations are sold all over the world, thus exposing cats and dogs of all ages and with various clinical conditions to its potential effects on gut microbial ecology and health. Moreover, prebiotics should theoretically increase the abundance of certain bacterial groups (e.g., *Lactobacillus* and *Bifidobacterium*) in the gut in order to be considered a prebiotic, given current definitions of these dietary compounds (*Gibson et al., 2010*). The objective of this study was to evaluate the effect of a commercially available product containing prebiotics on, fecal bacterial composition of clinically healthy cats and dogs. The results of this work show statistically significant differences in several bacterial groups that can be attributed to prebiotic administration. This study also lyprovides relevant insights into the uniqueness of baseline fecal bacterial populations and their highly individualized variability over time and response upon prebiotic administration in pet cats and dogs.

## METHODS

### Ethics

All experimental procedures were authorized by the Animal Care and Use Committee (AUP 2011–160) and the Clinical Research Review Committee at Texas A&M University (CRRC 10–14) and written informed client consent was obtained from the owners of all enrolled animals. Inclusion criteria included healthy (i.e., lack of clinical signs and good physical condition) non-obese, client-owned pet cats and dogs. Owners were instructed to feed their pets as usual without any supplement such as probiotics, prebiotics or vitamins. Exclusion criteria included abnormal serum parameters that could indicate subclinical abnormalities.

### Trial 1 (cats and dogs)

Clinically healthy client-owned and non-obese cats ($n = 12$) and dogs ($n = 12$) were enrolled (Table 1). Regardless of body weight, owners were instructed to feed 50 mL (containing 225 mg of FOS and inulin) of Viyo Veterinary® (proprietary mixture of vegetable and meat by-products, oils, vitamins and minerals containing 0.45% of prebiotics or 4,500 mg per kg in the end product) once per day for 16 days (this was the original dose recommended by the company). Although we deliberately did not control for the amount of food eaten per day, for a 10 kg dog eating 200 g of food per day this original dose would represent approximately 0.1% of dry matter intake. It should be noted that

**Table 1** Participant information (Trial 1, cats and dogs).

| Cats IDs | | | Final body weight |
|---|---|---|---|
| C2 | 4 years | DSH | 4.7 kg |
| C3 | 1 year 6 months | DSH | 6.2 kg |
| C4 | 6 years | DSH | 6.2 kg |
| C5 | 8 years | DSH | 5.2 kg |
| C8 | 2 years 6 months | Tabby | 4.2 kg |
| C10 | 4 years | Siamese mix | 5.1 kg |
| C11 | 5 years | DSH | 5.1 kg |
| C12 | 10 months | Siberian | 5.7 kg |
| C13 | 1 year 6 months | Calico | 2.8 kg |
| C14 | 1 year 2 months | DSH | 4.0 kg |

| Dogs IDs | Age | Breed | |
|---|---|---|---|
| D3 | 1 year 5 months | Doberman | 28.1 kg |
| D4 | 10 years | Rottweiler/Lab mix | 33.3 kg |
| D5 | 4 years | Boston Terrier | 10.5 kg |
| D6 | 1 year 6 months | Lab | 25.3 kg |
| D7 | 5 years | Lab mix | 23.6 kg |
| D8 | 4 years | Mixed | 23.1 kg |
| D9 | 7 years | Weimaraner | 29.1 kg |
| D10 | 1 year 10 months | Pembroke Welsh Corgi | 10.6 kg |
| D11 | 7 months | Mix hound/ Great Dane | 25.6 kg |
| D12 | 9 months | Australian Kelpie | 16.0 kg |

**Notes.**
DSH, Domestic Short Hair.

this prebiotic percentage of dry matter intake decreases proportionally to total dry matter intake. For example, for a 20 kg dog eating 400 g of food this original dose of 50 mL would only represent 0.06% prebiotic on a dry matter basis. Fecal samples were collected by the owners at two time points before prebiotic administration (8 days and 1 day before initiation of prebiotic administration) and again at two time points after initiation of prebiotic administration (days 8 and 16 after initiation of prebiotic administration) (see Fig. 1 for a timeline of our experimental design). Fecal samples were collected into special fecal sample tubes (provided), placed into zip-lock bags (provided) and frozen as soon as possible after collection. Samples were stored in the freezer until brought to our laboratory within 1–8 h, where they were stored at −20 °C until DNA extraction. The administration of 50 mL of Viyo Veterinary® daily for 16 days was the original dose recommended by the company in an effort to improve health by a modification in the gut microbiota (main objective of this current study).

## Trial 2 (dogs only)

Clinically healthy client-owned, non-obese dogs ($n = 10$) were enrolled (Table 2). Five of these dogs also participated in trial 1 (Trial 2 started approximately 9 months after Trial 1, therefore there is no risk on carryover effects). Owners were instructed to feed 3.2 ml/kg

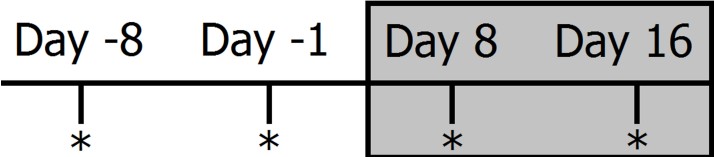

**Figure 1 Timeline of experimental design and sampling for 16S bacterial profiling (marked with \*).**
Two fecal samples were collected before (days −8 and −1) and during prebiotic administration (days 8 and 16). The prebiotic was administered daily to each animal for a period of 16 days (grey area).

**Table 2 Participant information (Trial 2, dogs only).**

| Dogs IDs | Age | Breed | Final body weight | Comments |
|---|---|---|---|---|
| D1 | 4 years 9 months | Boston Terrier | 10.5 kg | Same as D5 in Trial 1 |
| D3 | 8 years | Weimaraner | 29.5 kg | Same as D9 in Trial 1 |
| D4 | 11 years | Mix | 30.4 kg | Same as D4 in Trial 1 |
| D5 | 2 years 6 months | Doberman | 29.5 kg | Same as D3 in Trial 1 |
| D6 | 3 years 3 months | Mixed | 29.5 kg | New dog |
| D7 | 11 months | Dutch Shepherd | 20.4 kg | New dog |
| D8 | 9 months | Welsh Pembroke Corgi | 10 kg | New dog |
| D9 | 1 year 9 months | Australian Kelpie | 18 kg | Same as D12 in Trial 1 |
| D11 | 1 year 6 months | Australian Shepherd | 16.7 kg | New dog |
| D12 | 1 year 3 months | Pit Bull mix | 32 kg | New dog |

bodyweight (each mL containing 31 mg of FOS and inulin) of an especially formulated preparation of Viyo Veterinary® (containing 3.1% of prebiotics or 31,000 mg per kg in the end product) once per day for 16 days. The new formula was designed in an effort to reach high enough levels of prebiotics in the overall dry matter consumed that would be expected to have an impact on the intestinal microbiota in all dogs without reaching unfeasible amounts (in mL) of the product. For example, a 10 kg dog eating 200 g of food per day would need to consume 32 mL of the product (equating to 992 mg of prebiotics) and this new dose would represent approximately 0.5% of dry matter intake, while a 20 kg dog eating 400 g of food per day would need to consume 60 mL of the product (equating to 1,860 mg of prebiotics) and this new dose would also represent approximately 0.5% of dry matter intake. Similarly to trial 1, fecal samples were collected at two time points before prebiotic administration (8 days and 1 day before initiation of prebiotic administration) and at two time points after initiation of prebiotic administration (days 8 and 16 after initiation of prebiotic administration) (Fig. 1).

## Questionnaire

All pet owners (trials 1 and 2) were provided with a questionnaire to record the following parameters during the study period: acceptance of the prebiotic, attitude, appetite, drinking behavior, defecation frequency, borborygmus, flatulence, as well as volume, consistency,

and color of feces (Supplemental Information). This questionnaire has been used in other studies from our research group (*Rutz et al., 2004*).

## DNA extraction and 16S bacterial profiling

A bead-beating phenol-chloroform based-method was utilized to isolate total genomic DNA from all fecal samples as described elsewhere (*Suchodolski et al., 2005*). Primers specific for 16S rRNA genes were used to amplify the variable V4–V5 region as described previously (*Suchodolski et al., 2009*). Fecal bacterial communities were evaluated using 454-pyrosequencing before and during prebiotic administration using a bacterial tag-encoded FLX-titanium 16S rRNA gene amplicon pyrosequencing (bTEFAP) as described previously for canine and feline fecal samples (*Garcia-Mazcorro et al., 2011*; *Handl et al., 2011*). All sequences with their corresponding metadata information is freely available in the Sequence Read Archive at the NCBI (SRP071082).

## Sequence analysis

The open-source freely available bioinformatics pipeline Quantitative Insights into Microbial Ecology (QIIME) v. 1.8 was used to perform microbiome analysis from raw 16S DNA sequencing data using default scripts unless otherwise noted (*Caporaso et al., 2010*; *Navas-Molina et al., 2013*). The split_libraries.py was used to perform quality filtering and demultiplexing (i.e., assignment of reads to samples). Operational Taxonomic Units (OTUs) were assigned using two different approaches. First using UCLUST v.1.2.22 (*Edgar, 2010*) with an open reference script (pick_open_reference_otus.py, *Rideout et al., 2014*) in QIIME for alpha and beta diversity. Note that this algorithm does not necessarily discard sequences that do not match the reference 16S database, thus allowing for an accurate OTU representation. Second, using a closed reference algorithm (pick_closed_reference_OTUs.py) for further analysis using Phylogenetic Investigation of Communities by Reconstruction of Unobserved States (PICRUSt) (*Langille et al., 2013*). The GreenGenes 13_5 97% OTU representative 16S rRNA gene sequences was used as the reference sequence collection (*DeSantis et al., 2006*). Both weighted and unweighted UniFrac distances were used to investigate clustering of microbial communities (*Lozupone & Knight, 2005*; *Lozupone et al., 2007*).

## Statistical analysis

The linear discriminant analysis (LDA) effect size (LEfSe) method (*Segata et al., 2011*) was used to find organisms that could explain the differences in bacterial communities between the time periods before and during prebiotic administration. This method uses non-parametric tests and has been shown to be able to capture microbial taxa associated with class variables in several studies from our research groups (*Garcia-Mazcorro et al., 2016*; *Minamoto et al., 2015*). The ANOSIM and Adonis tests included in the compare_categories.py QIIME script were used to determine whether the grouping of samples (i.e., microbial communities) according to treatment period (i.e., before and during prebiotic administration) is statistically significant also in QIIME. An alpha of 0.05 was considered to reject null hypotheses.

## RESULTS

Viyo Veterinary® was well accepted (i.e., all except two cats in trial 1 and one dog in trial 2 had a good or excellent acceptance of the product at all time points during administration of the product, as perceived by the owners). No negative side effects from consuming the prebiotic preparation, such as vomiting, abdominal pain, lethargy, changes in fecal consistency, and/or diarrhea were reported by the owners. Briefly, 96% of all time points either before or during prebiotic in both trials were reported as normal or better than normal in all parameters measured that contained normal as a category. As perceived by the owners, one cat in trial 1 had lose or pulpy feces throughout the whole study period (i.e., before and during prebiotic), and one dog had some flatulence also throughout the study (in fact, this dog also participated in trial 2 and was also reported to present some flatulence during the whole study period). In trial 1 two cats refused to consume the product and were therefore excluded from the study. Also in trial 1, two dogs were excluded because of serum cobalamin and folate concentrations that were below the lower limit of the reference interval (1 dog) or microfilaria identified in the blood during the complete blood count (1 dog).

### Trial 1—cats

A total of 10 cats completed trial 1 (~4,600 quality-filtered sequences per sample; average 442 nucleotides per sequence) (Table 1). Similar to other studies (*Handl et al., 2011*), the fecal microbiota of cats was dominated by Firmicutes (median: 93.5%, range: 54.5–99.8%) followed by smaller proportions of Bacteroidetes (median: 3.4%, range: 0–37.1%) and other very low abundant groups (Fig. 2). Please note that each study reveals numbers and proportions of different microbial taxa that are the result of a combination of factors such as primers for 16S amplification, DNA extraction procedure, length of amplicon, reference sequence collection used to assign taxonomy, and inter-individual variability. The fecal microbiota in cats showed less intraindividual variability over time compared to interindividual variability (Fig. 3). Interestingly, cat number 2 (C2) and cat number 5 (C5) showed high increases in the relative abundance of Lactobacillales (mostly *Lactobacillus* spp.) during prebiotic administration (Fig. 3), which is noteworthy given the historical association of prebiotics with increased abundances of lactic acid bacteria.

The LEfSe method showed that an unknown member of the family Veillonellaceae (order Clostridiales within Firmicutes) was significantly increased during prebiotic administration, and also that an unknown member of Gammaproteobacteria was decreased during prebiotic administration (Fig. 4). These changes, which involved samples from several individual cats thus suggesting an effect of the prebiotic (Fig. 4), were not sufficient to cause a significant difference in bacterial communities using weighted UniFrac distances (Fig. 5, please note that the analysis of unweighted UniFrac distances revealed similar results). This lack of significance is supported by high $p$-values in ANOSIM and Adonis tests ($p > 0.5$), even though these tests are known to have very low specificity (i.e., these tests usually detect a difference in microbial communities even when there is not necessarily a strong and clear separation in PCoA plots). A predictive approach to investigate the functional microbiome

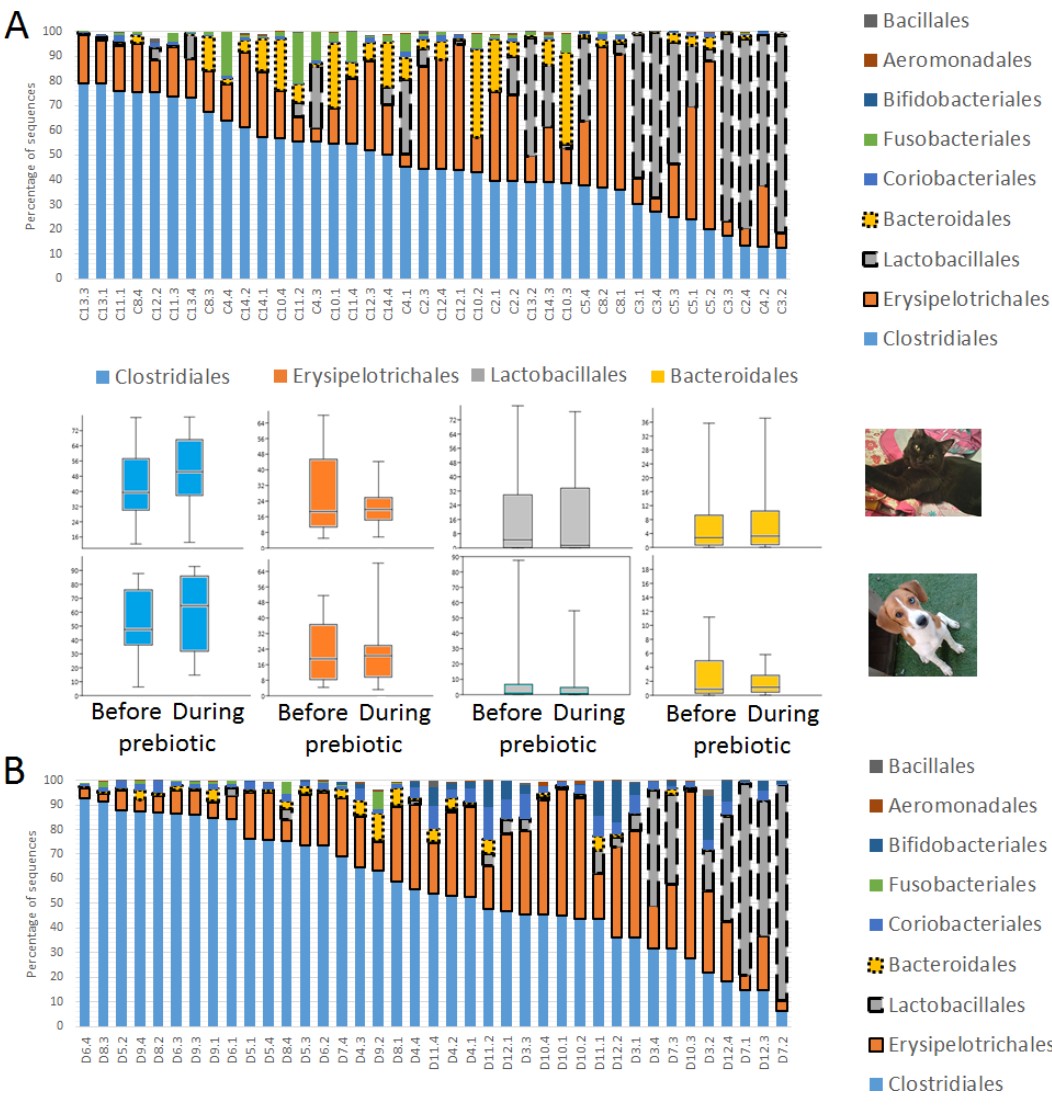

**Figure 2** **Relative abundance of bacterial groups at the order level in trial 1.** This figure displays column charts that show the relative abundance of 16S sequences at the order level for cats (A) and dogs (B). Samples were organized based on the highest abundant order (Clostridiales). Box plots for the most abundant orders are also shown (most bacterial groups did not show a statistical significant difference; see main text for details). The x axis contains the sample names (C, cats; D, dogs, numbers imply the number of the animal and the day of sampling (1, day −8, 2, day −1, 3, day 8; 4, day 16). For example, C13.3 implies cat number 13, day 8 during prebiotic administration.

using PICRUSt did not reveal any significant difference between the period before and during prebiotic administration (Supplemental Information).

## Trial 1—dogs

A total of 10 dogs completed trial 1 (~4,600 quality-filtered sequences per sample; average 442 nucleotides per sequence). One sample (dog number 11 or D11, day 8 during prebiotic administration) did not produce any sequence data and could not be used for analysis. Similar to other studies (*Handl et al., 2011*), the fecal microbiota of dogs was

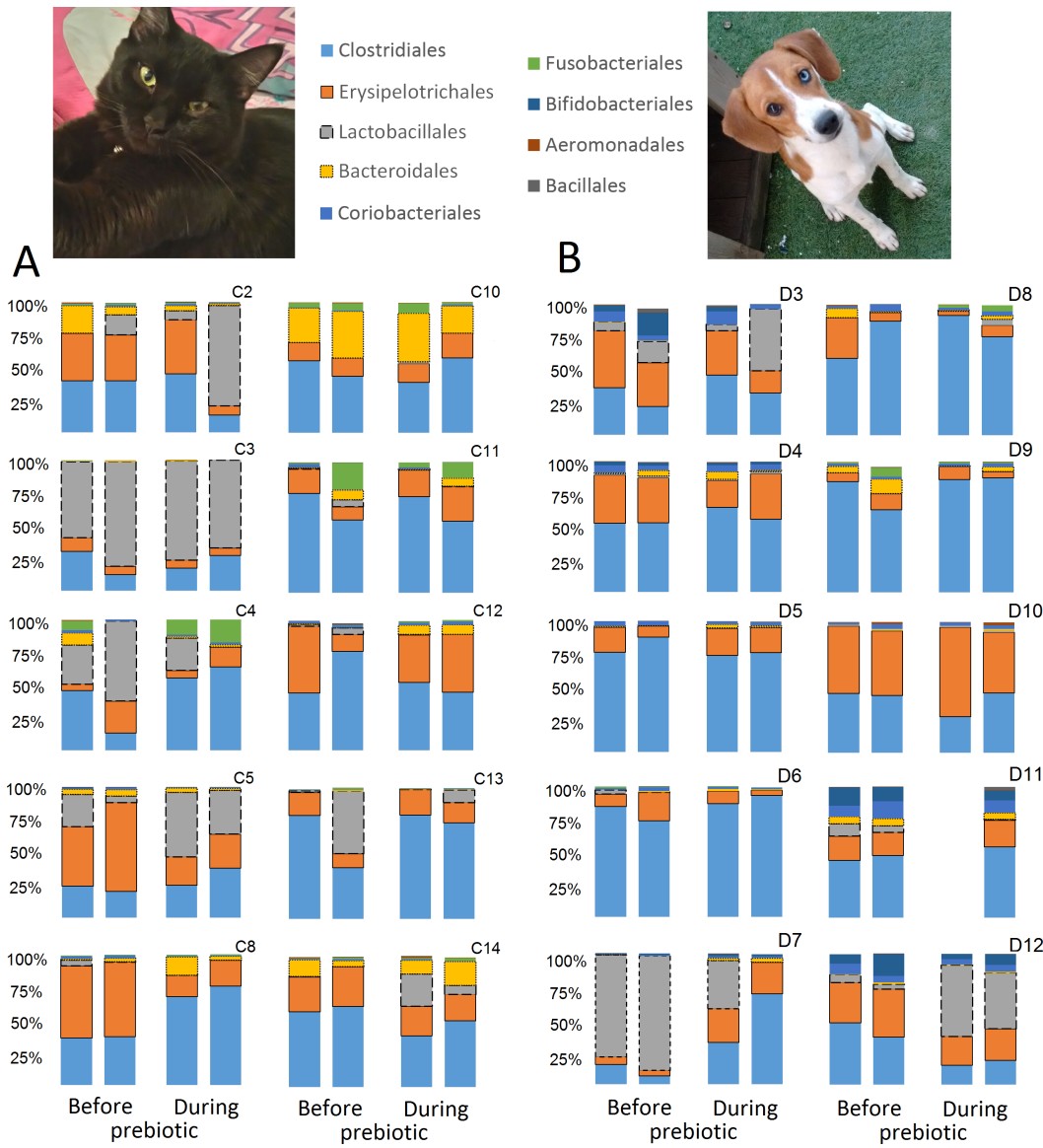

**Figure 3** **Relative abundance of bacterial groups at the order level for each cat and dog in trial 1.** The sample names for cats (C) and dogs (D) are numbered depending on the animal ID (see Table 1). Bars represent day 8 and 1 before prebiotic day 8 and 16 during prebiotic administration, in that order. Please note that a sample corresponding to day 8 during prebiotic administration in Dog 11 (D11) could not be analyzed.

dominated by Firmicutes (median: 93.2%, range: 70.2–98.8%) (Fig. 2) with each dog also having unique patterns of fecal bacterial abundances showing stability over time (Fig. 3). Two dogs (dog number 3, D3, and dog number 12, D12) showed high increases in the order Lactobacillales during prebiotic administration (Fig. 3) although D12 did not show the same increase in this bacterial group in trial 2 (see Trial 2 below). Interestingly, another dog in trial 1 (dog number 7, D7) had very high abundances of Lactobacillales at baseline (before prebiotic administration) and these abundances decreased to near

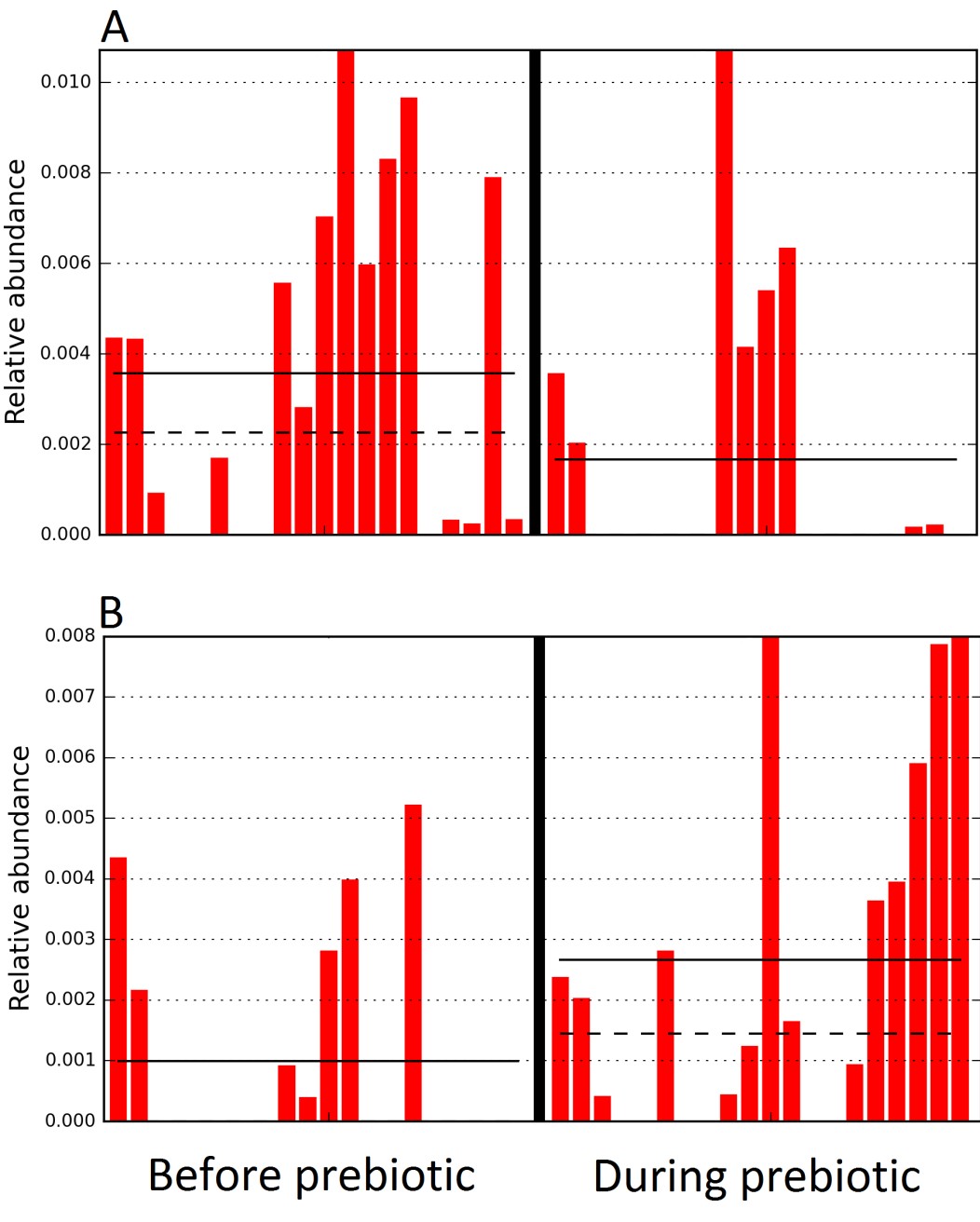

**Before prebiotic**      **During prebiotic**

**Figure 4** **Relative abundance of bacteria in cats in trial 1 before and during prebiotic administration.** The LEfSe method revealed a significant difference in the relative abundance of Gammaproteobacteria (A) and Veillonellaceae (B) between the periods before and during prebiotic administration. Straight lines represent medians and dashed lines represent means.

0% at day 16 of prebiotic administration (Fig. 3). The LEfSe method showed that an unknown member of Staphylococcaceae was higher during prebiotic administration, while the genus *Sutterella* (family Alcaligenaceae in the order Burkholderiales within the Betaproteobacteria) was higher (although less prevalent) before prebiotic administration (Fig. 6). It is important to note that these differences were due to a few samples only

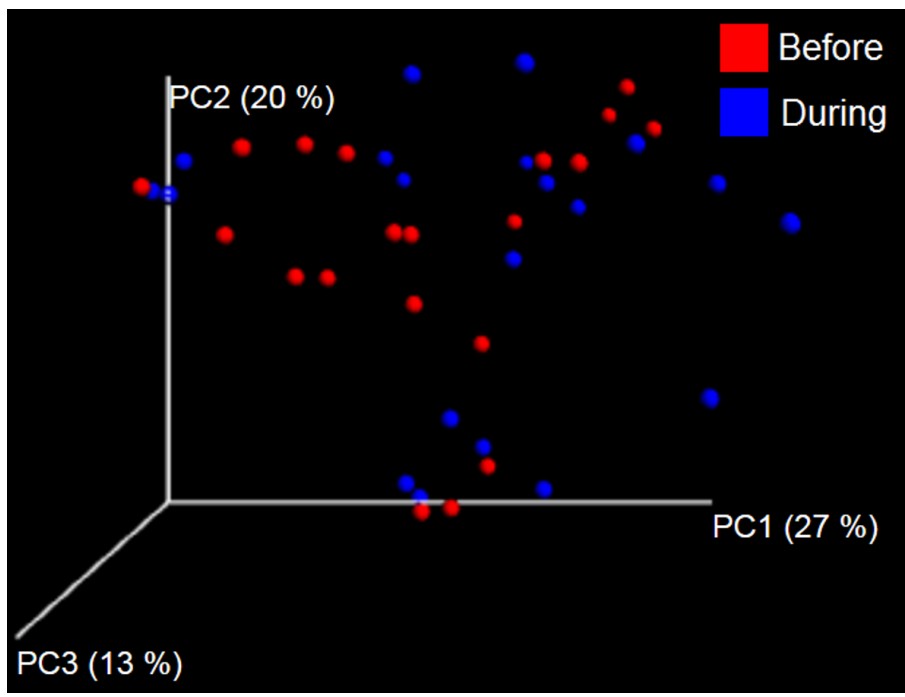

**Figure 5** **Principal Coordinate Analysis (PCoA) plot.** PCoA plot of weighted UniFrac distances in cats (trial 1). The lack of clustering by treatment was supported by ANOSIM and Adonis tests ($p > 0.5$, see main text).

(especially for Staphylococcaceae), which were nonetheless enough for the LEfSe method to detect a significant effect (Fig. 6). Similarly to cats, these differences were not sufficient to significantly separate bacterial communities according to weighted UniFrac distances (Fig. 7, unweighted UniFrac revealed similar results). Also similar to what was observed in cats, a predictive approach to investigate the functional microbiome did not reveal any significant difference between the period before and during prebiotic administration in the dogs enrolled in trial 1 (Supplemental Information).

## Trial 2—dogs only

Trial 2 was designed to explore the possibility that an increase in prebiotic content would result in relevant changes in the fecal microbiota, with a focus on canine patients. A total of 10 dogs completed trial 2 (~3,200 quality-filtered sequences per sample; average 432 nucleotides per sequence). The fecal microbiota was again dominated by Firmicutes although with much lower proportions compared to all dogs in trial 1 (median: 78.5% in trial 2 vs. a median: 93.2% in trial 1) and with higher variability (range: 29.6–97.6% vs. 70.2–98.8% in trial 1) (Fig. 8). The reasons behind these differences in relative proportions and variability in the phylum Firmicutes (and other bacterial groups) are unclear; for example, 5 dogs participated in both trials but these dogs showed bacterial abundances and over time variability (Fig. 9) that did not necessarily reflect those abundances and variability in trial 1 (Fig. 3). Actually, dog 12 (D12) participated in both trials but only showed increases in Lactobacillales in trial 1 (this dog was coded as dog number 9 or D9 in

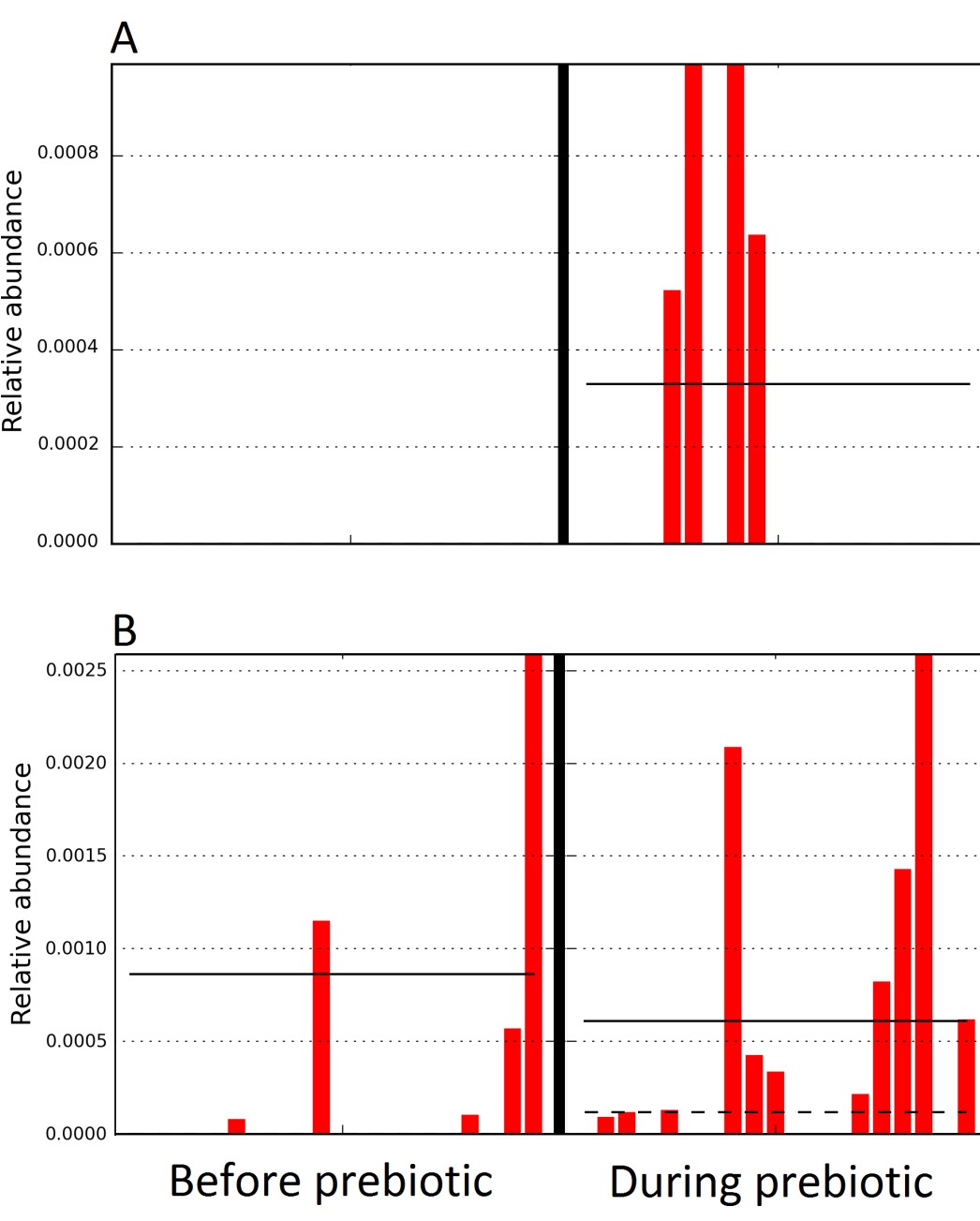

**Figure 6  Relative abundance of bacteria in dogs in trial 1 before and during prebiotic administration.** The LEfSe method revealed a significant difference in the relative abundance of Staphylococcacea (A) and *Sutterella* (B) between the periods before and during prebiotic administration. Straight lines represent medians and dashed lines represent means.

trial 2). Interestingly, one dog in trial 2 (dog number 7, D7) had near 0% *Bifidobacterium* at both time points before prebiotic administration, an increase to 8.4% on day 8 after initiation of prebiotic administration, and a further increase to 25.9% on day 16 after initiation of prebiotic administration (Fig. 9). This same dog (D7, trial 2) also had a massive increase of Lactobacillaceae from <1% before and on day 8 after initiation of prebiotic

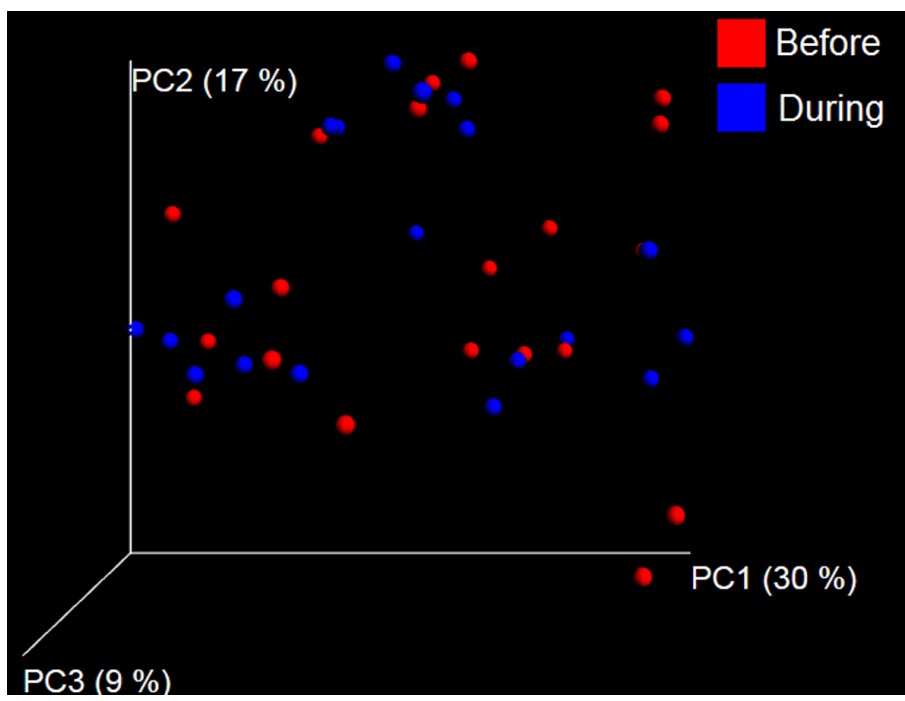

**Figure 7 Principal Coordinate Analysis (PCoA) plot.** PCoA plot of weighted UniFrac distances in dogs (trial 1). The lack of clustering by treatment was supported by ANOSIM and Adonis tests ($p > 0.5$, see main text).

administration to 35.2% on day 16 after initiation, and Turicibacteraceae, from 0% before prebiotic administration to 49% and 15% on days 8 and 16 after initiation of prebiotic administration, respectively (Fig. 9).The LEfSe method showed a lower abundance of *Dorea* (family Clostridiaceae) and also higher abundances of *Megamonas* and other (unknown) members of Veillonellaceae (class Negativicutes within the Firmicutes) during prebiotic administration (Fig. 10). These changes involved samples from several individuals and can therefore be considered associated with the prebiotic administration tested in trial 2 (Fig. 10). Similar to dogs in trial 1, these differences were not sufficient to significantly separate bacterial communities according to weighted or unweighted UniFrac distances, or to cause significant differences in the predicted functional microbiome (Supplemental Information).

## DISCUSSION

Prebiotics are non-digestible dietary ingredients with suggested health-bearing properties that are included in several commercially available products for use in cats and dogs. Sound scientific evidence shows that prebiotics can exert a positive effect on vertebrate (including humans) health (*Montalban-Arques et al., 2015*), but this has not been well studied in veterinary medicine, especially with regards to products that are commercially available. This study evaluated the fecal bacterial microbiota in healthy cats and dogs that were supplemented with a commercial prebiotic formulation containing FOS and inulin.

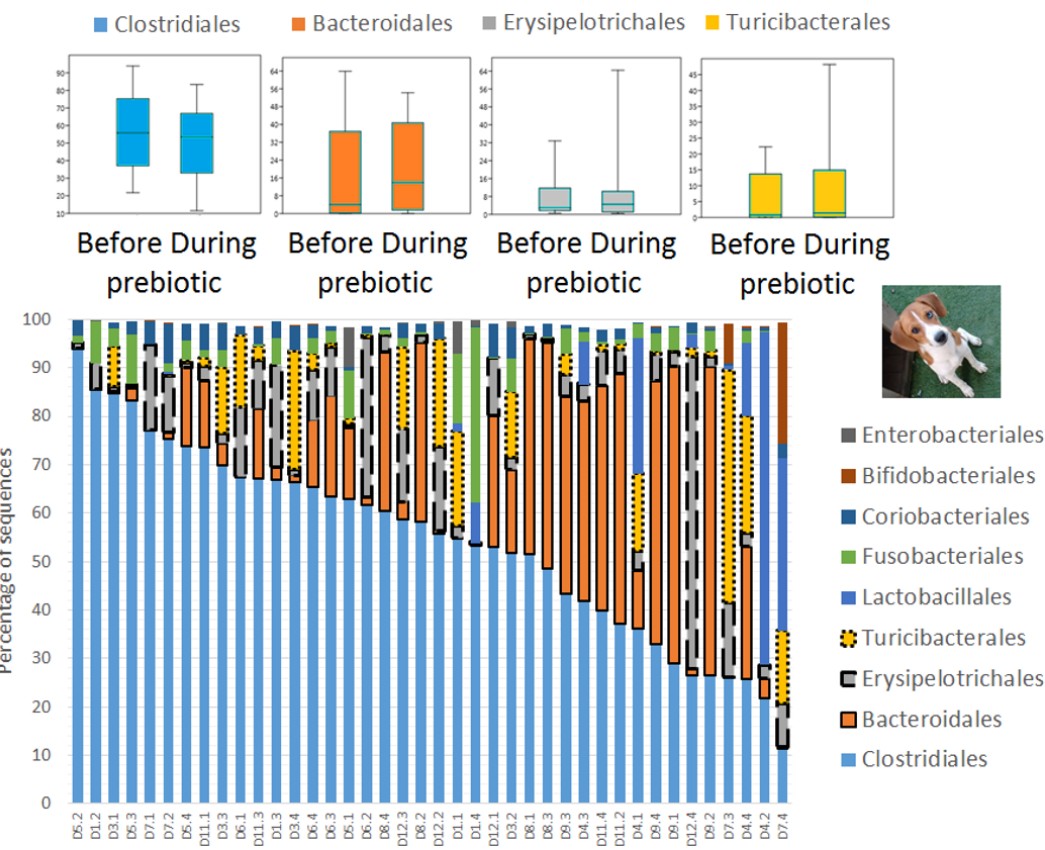

**Figure 8  Relative abundance of bacterial groups at the order level in trial 2.** This figure displays column charts that show the relative abundance of sequences at the order level for dogs (trial 2). Samples were organized based on the highest abundant order (Clostridiales). Box plots for the most abundant orders are also shown (most bacterial groups did not show a statistically significant difference; see main text for details). The *x* axis contains the sample names (D, dog; numbers imply the number of the animal and the day of sampling (1, day −8; 2, day −1; 3, day 8; 4, day 16)). For example, D5.2 implies dog number 5 day -1 before prebiotic administration.

    Our results support the fact that each individual animal (including humans) carries a microbial community so specific that it resembles a fingerprint (*Suchodolski et al., 2005*; *Zoetendal, Akkermans & De Vos, 1998*). In fact, research performed on the human microbiota has demonstrated the feasibility of microbiome-based identifiability of single individuals (*Franzosa et al., 2015*). While the factors associated with this uniqueness are a matter of debate, several studies in humans have shown that host genetics exert a great influence (*Benson et al., 2010*; *Blekhman et al., 2015*), although diet may indeed outweigh the effects of host genetic background (*Dabrowska & Witkiewicz, 2016*; *Wu et al., 2011*). This microbial uniqueness is particularly important to clinicians (both human and veterinary) because it also implies individualized responses to treatment (*Topol, 2014*), for example to antibiotic administration (*Dethlefsen & Relman, 2011*; *Igarashi et al., 2014*; *Suchodolski et al., 2009*). Unfortunately, guidelines for prebiotic administration are often unclear (i.e., companies usually suggest the same dose regardless of body weight, age,

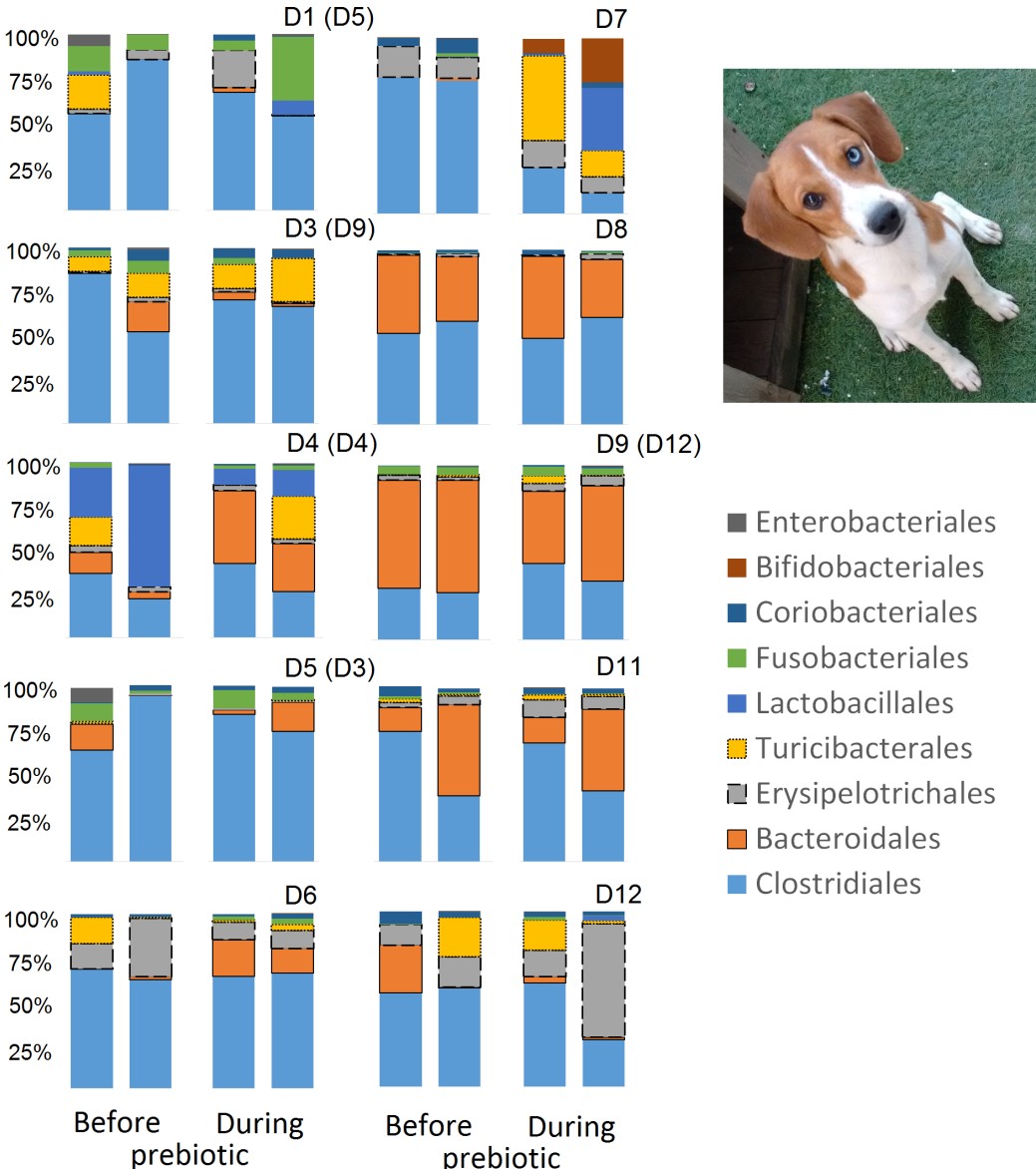

**Figure 9 Relative abundance of bacterial groups at the order level for each dog in trial 2.** The sample names are numbered depending on the animal ID (see Table 1). Within parenthesis, we also included the dog's ID based on trial 1. Bars represent day 8 and 1 before prebiotic and day 8 and 16 during prebiotic administration, in that order.

clinical condition, etc.) and have not fully considered the uniqueness of each gut microbial ecosystem (*Barzegari & Saei, 2012*).

While the individuality of gut microbial communities with regard to their response to prebiotic administration is a relevant matter for daily clinical use of these increasingly utilized nutraceutical ingredients, very few studies have discussed the uniqueness of native bacterial communities in individual cats or dogs, their variability over time or during the course of particular treatments (*Garcia-Mazcorro et al., 2012a*; *Garcia-Mazcorro et al., 2012b*; *Ritchie, Steiner & Suchodolski, 2008*; *Suchodolski et al., 2005*). In this study, two

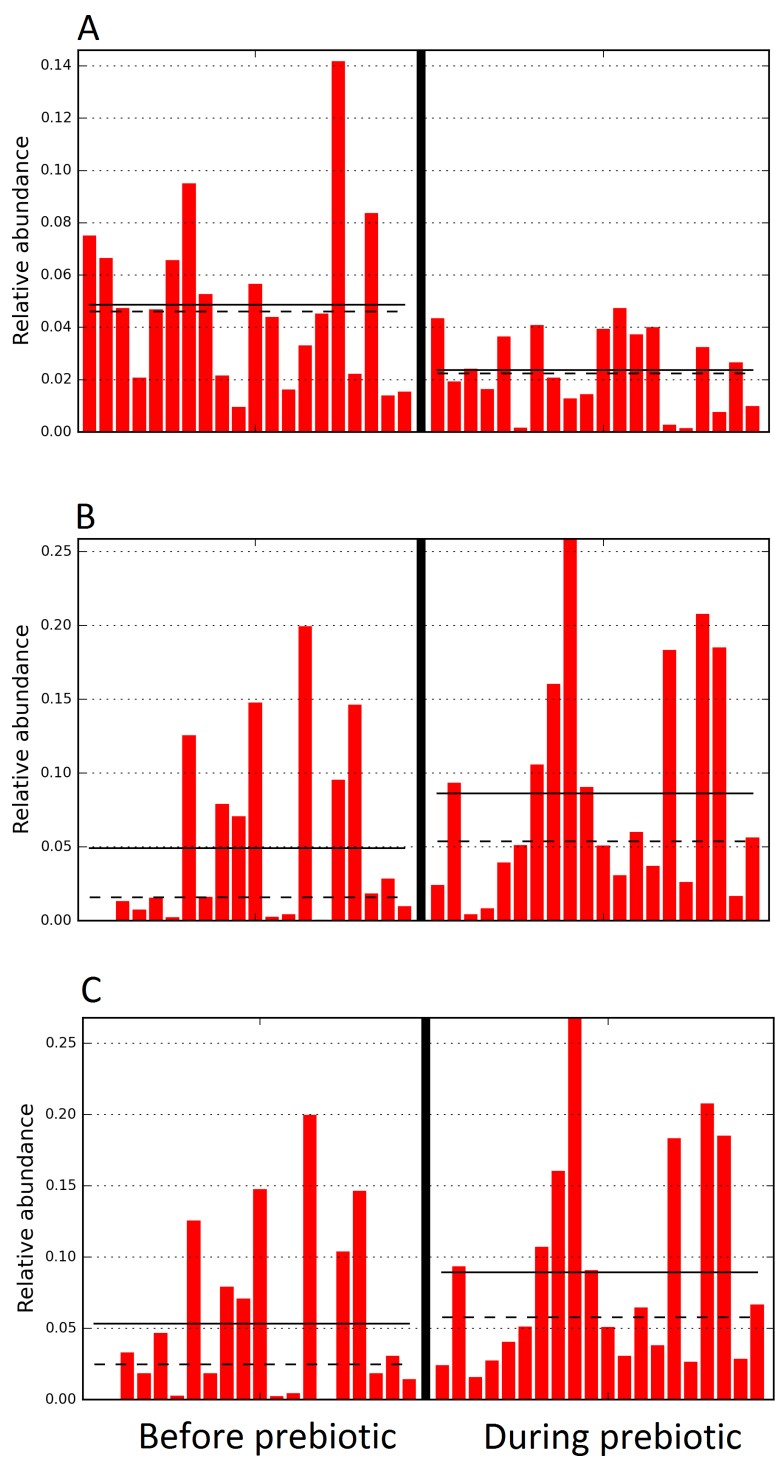

**Figure 10  Relative abundance of bacteria in dogs in trial 2 before and during prebiotic administration.**
The LEfSe method revealed a significant difference in the relative abundance of *Dorea* (A), *Megamonas* (B), and Veillonellacea (C) between the periods before and during prebiotic administration. Straight lines represent medians and dashed lines represent means.
cats and two dogs showed increases in the relative abundance of Lactobacillales (Fig. 3), suggesting that these animals are highly responsive individuals to the prebiotic tested (at least with regards to Lactobacillaes). Also in trial 1, one dog had very high abundances of Lactobacillales at baseline (i.e., before prebiotic) but showed a marked decrease to near 0% at day 16 of prebiotic administration (D7, Fig. 3). While these results suggest a relationship between baseline bacterial populations and response to prebiotics, this phenomenon has received very little attention (*Garcia-Mazcorro et al., 2011*; *Stecher et al., 2010*; *Arciero et al., 2010*; *Vitali et al., 2009*). For example, *Vitali et al. (2009)* mention that the significant increase of a bacterial group (i.e., *Lactobacillus helveticus*) after administration of a synbiotic (containing fructo-oligosaccharides, *L. helveticus* and *Bifidobacterium longum*) was directly linked to the low incidence of this group in the intestine of the human host, thus implying a potential relationship between the native bacterial groups and any other group that is being supplemented in the form of probiotics or that increases due to the presence of prebiotics. In support of this hypothesis, one dog in trial 2 (dog number 7, D7) had near 0% *Bifidobacterium* at baseline before prebiotic administration but showed a remarkable increase of this bacterial group on day 16 after initiation of prebiotic administration (Fig. 9), suggesting that this dog may also be considered a highly responsive individual to the prebiotic tested. Overall, our results support the concept that the native microbiota in each individual cat or dog is unique and that this microbiota show highly individualized patterns of variation over time and during the course of prebiotic administration.

In addition to confirming the uniqueness of fecal microbiota in individual cats and dogs, this study also confirms previous observations about the minimal effects of low prebiotic dosages on the gut microbiota of healthy cats (*Sparkes et al., 1998*; *Kanakupt et al., 2011*) and dogs (*Willard et al., 1994*; *Willard et al., 2000*; *Barry et al., 2009*; *Vanhoutte et al., 2005*). This minimal effect of prebiotics on the gut microbial ecosystem is a common result when administering low doses of prebiotics (~1% of dry matter) but higher doses have shown a potential to promote a more generalized effect of these ingredients in both cats (*Barry et al., 2010*) and dogs (*Middelbos et al., 2010*). However, conflicting results have been presented in the literature where different amounts of dietary fiber (0.5, 1, 2, 4, and 8%) were not associated with differences in the abundance of different microorganisms (*Faber et al., 2011*). Nonetheless, these and other similar studies often lack sufficient representativeness with regard to complex microbiota (i.e., most reports only studied one or a few organisms while hundreds of different microorganisms exist and cohabit the intestinal tract of cats and dogs) and some only used culture techniques, which are considered obsolete in contemporary studies of microbial ecology (*Ritz, 2007*). Indeed, studies such as this current investigation that uses high-throughput sequencing allows us to investigate the majority of all bacterial groups at once, thus offering valuable insights to current prebiotic literature in small veterinary practice.

The dose of prebiotics offered to each individual patient is a matter of debate in human and veterinary medicine. There are at least three possible ways to administer prebiotics to cats and dogs in real life. First, prebiotics can be offered as a fixed percentage of dry matter. Indeed, most well-controlled prebiotic papers in cats and dogs report the dose of prebiotics in percentage of dry matter intake, varying from 0% to 7% (*Patra, 2011*). A

potential issue with this way of administering prebiotics (i.e., as percentage of dry matter) is that the amount of food consumed by a given pet cat or dog may vary substantially over time (e.g., accordingly to age) and among different animals (e.g., two dogs, each weighting 10 kg, may consume different amounts of food). Therefore, in a real-life scenario (not a controlled setting) two individual animals having the same body weight may consume different amounts of total prebiotics in their diets, not because of the prebiotic percentage of dry matter but because of the different amounts of food consumed. Second, a fixed amount of prebiotics can be offered regardless of dry matter intake, age, body weight, and all other specific characteristics of the animal. For example, 50 mL were offered in trial 1 regardless of body weight and the amount of food consumed each day to each cat and dog (please note that this was original dose provided by the company). This dose has the disadvantage that the amount of prebiotic offered would decrease proportionally to the amount of food consumed. Finally, another way of administering prebiotics can be based on other parameters aside dry matter intake. For example, trial 2 was designed to equilibrate the amount of prebiotics for each dog, using a straightforward parameter (i.e., body weight). Interestingly, in this current study we report increases in important bacterial groups for gut health such as Veillonellacea (*Suchodolski et al., 2012*) only in trial 2. This discussion and the data generated by this current study may be relevant to guide other studies addressing the effect of products containing prebiotics offered to cats and dogs.

Our study evaluated a product that, together with other prebiotic formulations, are currently marketed to all breeds of cats and dogs of all ages, sizes and clinical conditions. Therefore, our study adds relevant information for the potential effect of commercial prebiotics. Nonetheless, there are at least five potential limitations of this study that are important to discuss for guiding future efforts in using prebiotics to improve intestinal health in cats and dogs. First, in this study we included a highly diverse group of animals, which may have influenced the response or lack thereof to prebiotic administration. The inclusion of a more homogeneous group of animals may have diminished this variability and therefore make the effect of prebiotic administration easier to detect. However, this is not always the case. For example, a recent study showed a minimal effect of potato fiber on the fecal microbiota of dogs using a homogeneous group of animals (all female with hound bloodlines and similar age and body weight) (*Panasevich et al., 2015*). In this study we deliberately included different dogs to mimic a real life scenario. Second, in this study we deliberately did not force the owners to feed a specific amount of food per day but instead we asked them to continue their regular feeding habits. This is important to investigate native microbial communities and their fluctuations in ordinary pets, which are the ultimate consumers of nutraceuticals containing prebiotics. Third, in this study we only used one molecular technique (i.e., high-throughput sequencing) to assess the fecal microbiota, and other studies have shown that the results from this technique do not always correlate with the results of other molecular techniques such as fluorescent *in situ* hybridization (*Garcia-Mazcorro et al., 2012a*; *Garcia-Mazcorro et al., 2012b*). Nonetheless, other studies have shown that sequencing results correlate well with the results obtained from other molecular techniques such as quantitative real-time PCR (*Minamoto et al., 2015*; *Panasevich et al., 2015*). Fourth, commercial prebiotic formulations such as the one

used in this study contains a mixture of ingredients aside the prebiotic component that makes it difficult to study the effect of the prebiotics independently. Lastly, in this study we only evaluated the bacterial microbiota but the fungal microbiota does indeed deserve investigation (*Handl et al., 2011*).

In summary, there is a potential beneficial effect of prebiotics to improve gut health in cats and dogs and this effect may be mediated by changes in the gut microbiota (*Schmitz & Suchodolski, 2016*). This study reinforces the notion that individual cats and dogs have a unique fecal microbiota, which is relatively stable over time and responds differently to dietary manipulation using prebiotics and possibly other dietary compounds. Also, this study shows that the consumption of up to 31 mg/kg body weight of prebiotics (a mixture of FOS and inulin) does not significantly change the abundance of most bacterial groups in feces of healthy dogs. Exceptions include bacterial groups such as *Dorea*, *Megamonas*, *Sutterela*, Veilloneceae, Staphylococcaceae, and Gammaproteobacteria, which deserve attention because the changes observed in this study (although largely driven by individual responses) were not accompanied by negative side effects. Veillonellaceae deserves particular attention because it showed increased abundances during prebiotic administration in cats (trial 1) and dogs (trial 2) in this current study and other studies have shown that this group is depleted in the duodenum of dogs with idiopathic inflammatory bowel disease (*Suchodolski et al., 2012*) and is highly responsive to dietary challenges (*Bonder et al., 2016*), including consumption of soluble corn fiber and polydextrose in humans (*Hooda et al., 2012*) and inulin in dogs (*Beloshapka et al., 2013*). Importantly, this study was not performed in a controlled setting; therefore controlled studies with control of diet, environment and individual characteristics of the animals such as breed and age, may help to draw more conclusive evidence about the effect of prebiotics on the gut microbiota of pet cats and dogs. Our current study does not rule out other mechanisms by which the evaluated product may confer a health benefit to the host (e.g., increased production of short-chain fatty acids), but more studies are needed to prove this and to study in more detail the effect of this and other commercially available products containing prebiotics for cats and dogs. Moreover, more studies are needed to explore potentially beneficial effects on host health beyond changes in bacterial communities such as increased expression of immunoregulators in the intestinal mucosa (e.g., cytokines).

### Funding
This study was made possible by the financial support from Viyo International (http://www.viyo.com/). The funders had no role in study design, data collection and analysis, decision to publish, or preparation of the manuscript.

### Grant Disclosures
The following grant information was disclosed by the authors:
Viyo International.

## Competing Interests

The authors declare there are no competing interests.

## Author Contributions

- Jose F. Garcia-Mazcorro conceived and designed the experiments, performed the experiments, analyzed the data, wrote the paper, prepared figures and/or tables, reviewed drafts of the paper.
- Jose R. Barcenas-Walls analyzed the data, wrote the paper, reviewed drafts of the paper.
- Jan S. Suchodolski conceived and designed the experiments, performed the experiments, analyzed the data, contributed reagents/materials/analysis tools, wrote the paper, reviewed drafts of the paper.
- Jörg M. Steiner conceived and designed the experiments, performed the experiments, contributed reagents/materials/analysis tools, wrote the paper, reviewed drafts of the paper.

## Animal Ethics

The following information was supplied relating to ethical approvals (i.e., approving body and any reference numbers):

All experimental procedures were authorized by the Animal Care and Use Committee (AUP 2011-160) and the Clinical Research Review Committee at Texas A&M University (CRRC 10-14) and written informed client consent was obtained from the owners of all enrolled animals.

## DNA Deposition

The following information was supplied regarding the deposition of DNA sequences:
Sequence Read Archive at the NCBI (SRP071082).

## Data Availability

The raw data is available at the Sequence Read Archive at the NCBI (SRP071082).

## Supplemental Information

Supplemental information for this article can be found online at http://dx.doi.org/10.7717/peerj.3184#supplemental-information.

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
