# Peer review of "Molecular assessment of the fecal microbiota in healthy cats and dogs before and during supplementation with fructo-oligosaccharides (FOS) and inulin using high-throughput 454-pyrosequencing"

_PeerJ, doi:10.7717/peerj.3184_

## Round 0.1 · original submission · Major Revisions

· Academic Editor

Major Revisions

The manuscript should be revised to reflect that these data represent a case series. Since the selection of the study subjects and administration of prebiotic was not done in a controlled manner, the data are most useful for hypothesis generation. The application of statistical comparisons is not particularly informative given the nature of the study. In addition, the statistical analysis section is lacking necessary details, such as definition of statistical significance, description of tests done and why they were applied here. The manuscript should be revised to reflect the exploratory nature of the study, emphasize the finding of individual variation, and describe how further studies can build on these findings to evaluate whether prebiotics do indeed have a measureable effect on the microbiome. In addition, clinical relevance needs to be addressed (i.e. include summary of questionnaire data that might shed light on effect of prebiotic on appetite, elimination, etc..). A summary of the patient weight data, along with the estimated dose for trial 1, should be also provided to permit an evaluation of how the doses may have differed between the two trials.

Reviewer 1 ·

Basic reporting

Overall the article has been written clearly with unambiguous text. Relevant prior literature has been referenced; however, the article would benefit from a more detailed description of the literature referenced. For example:

• Lines 71-78 of the introduction mention that many articles have been published showing the extent of microbial symbiosis in health and during disease states and that these studies are highly supported by high-throughput sequencing technologies; however, no specific evidence of this is offered.

• Lines 85-86 indicate prebiotics influence distant sites such as bones and skin but do not mention what types of effects are seen or whether these effects are positive or negative.

• Lines 87-89 mention that hundreds of research studies have shown beneficial effects associated with the consumption of fiber on gut microbiota; however, these beneficial effects are not defined. In addition, no specific study mentioning a beneficial effect is referenced.

• The figures may benefit from having samples from each participant animal grouped together so that it is easier to see changes over time and with prebiotic administration.

The text in this article gets the author’s points across; however, it would benefit from more careful review of grammar and syntax. The Discussion, Caveats and Summary sections could potentially be combined into one or two sections rather than three.

Experimental design

The research question, of how a commercially available prebiotic, effects fecal bacterial composition of clinically healthy dogs and cats has been well defined and clearly stated. The knowledge gap being investigated is identifiable. The information provided for each individual trial is clear and reproducible. Some concerns regarding the reproducibility of the study outside of the individual trial parameters are as follows:

• The sequence analysis section (lines 167-177) requires a more detailed description of the methodology. For example, simply saying several tools in QIIME were used for quality filtering is not sufficient for reproducibility. References were supplied; however, a more detailed explanation of tools used, workflow followed and any changes to stock parameters are necessary.

• In the statistical analysis section (lines 179-185) there is no explanation of how the ANOSIM and Adonis tests were used to compare microbial communities. The first sentence of this section (line 180) could be better elaborated upon, what specific differences are being measured?

• In the results it is mentioned that 2 dogs were excluded due to serum cobamin and folate concentrations being below the reference limit; however, I did not see any mention of exclusion criteria or the testing methods for these parameters in the methods section.

• In Trial 2 (dogs only) a new formula was given to the dogs in an effort to reach high enough levels of prebiotics in overall dry matter consumed that would be expected to have an impact on the intestinal microbiotia in all dogs; however, there is no indication of how this level was determined.

• Weighted UniFrac distances are mentioned as the method used detect separation of bacterial communities in the results section but no mention of weighted UniFrac distances is made in the methods section.

• It is curious that the animals were fed Viyo for 24 days and yet were only tested on days -8, -1, +8 and +16 not day 24. What is the reason for these time points?

• 5 dogs were included in both Trial 1 and Trial 2. The length of time between trials was not indicated, I wonder if the results of Trial 1 could have influenced the results of Trial 2 in those dogs that participated in both.

Validity of the findings

As Viyo Veterinary appears to contain elements not described in the paper such as vitamins, minerals, fatty acids and amino acids in addition to the prebiotics how can the authors confidently conclude that the changes in intestinal microbiota were due to the prebiotics rather than other components of the product? It does not appear that the effect of other components of the product were controlled for in this study. In order to draw meaningful conclusions about prebiotics these elements would need to be controlled for.

In the discussion (line 307-309) a relationship between baseline bacterial populations and response to prebiotics has been suggested; however, evidence of this was not mentioned in the results section. The discussion would benefit from evidence showing how the authors arrived at this conclusion.

Additional factors that may influence the validity of this study, such as no control over diets, are addressed by the authors in the caveats section. Although the authors are trying to simulate a real life situation I believe the effects need to be studied in a carefully controlled laboratory setting first. There are to many other variables at play to make any conclusions on the effect of prebiotics. A higher number of test subjects would have benefited the study as it is difficult to draw conclusions based up on a sample number of 10 with no repeated experiments to confirm results.

In the summary the authors indicate that prebiotics have a beneficial effect on the animals in the study; however, no where do they prove that there is a measurable effect (either positive or negative) on gut health.

Additional comments

Your study is interesting and provides useful information to the veterinary community about the use of prebiotics; however, I believe that this study needs to be done in a controlled laboratory setting prior to attempting to simulate real life situations. If you did this study as mentioned above and added in your data simulating real life scenarios I believe it would be a more complete paper. It is very difficult to draw conclusions about prebiotics when the presence of other ingredients in Viyo are not controlled for in any way.

Reviewer 2 ·

Basic reporting

This manuscript meets the standards of this journal.

Experimental design

This manuscript meets the standards of this journal, however further clarification is needed prior to its publication.

Validity of the findings

Despite the lack of a robust response due to prebiotic supplementation, this manuscript discusses valid points related to the challenges of studying the use of prebiotics in canine and feline nutrition. In addition, little research is available on this topic, which adds further value of the results and points discussed in this manuscript.

Additional comments

Dear authors:

General comments:
This is a well-written manuscript that should be considered for publication in this journal after a few edits and further clarifications about its content. The first half of the discussion section should be condensed, as a detailed discussion of individual animal response is not always relevant, especially when they are variable and not related to the treatment applied. In addition, information about how far apart trial 1 and 2 were from each other is needed, as well as the potential for carryover effect on the 5 dogs that had been previously exposed to the prebiotic.

Further clarifications needed: 1. How were the fecal samples collected and stored until DNA extraction. 2. Was the questionnaire used validated/ previously used? 3. What were the inclusion and exclusion criteria for trial 1 and 2? Did you exclude animals that were being fed diets or supplements that were either pre or probiotics?

Specific comments:

Title: change molecular technique to a more specific term for the technique you used (e.g., 454 pyrosequencing assessment….)

Keywords: best to use “feline” and “canine”, instead of “dogs” and “cats” to expand the reach of this manuscript during a literature search. As the two latter terms already show on the title.

Line 27-29: needs to be rewritten.

Line 37: what side effects were expected? If space permits, please add few examples.

Line 43: “predictive approach” was this PICRUSt? If so, please add info.

Line 45-46: needs to clarify that this was a separate study. It was not clear until reading the material and methods.

Line 57-58: what would you suggest as further measurements? This statement does not help to readers to understand how this field can be advanced.

Line 70: change “massive” to something more quantitative.

Lines 82-84: not all fibers are fermentable. Thus, this statement is not correct as written, nor all are prebiotics. This sentence needs to be revised.

Line 86: “influence” what does it mean in this context? Please be more specific about how microbiota affect other/ distant body sites.


Line 98: Not sure what the authors mean by raw prebiotics. Several prebiotic sources have been through an extraction process, other fiber sources are added in pet foods that then are processed. Thus, further clarification is needed in this statement.

Line 119: what was the BW range and age of the animals selected for this trial?

Line 130: Why were feces collected at 8 and 16 days post supplementation, but not at day 24? If this is correct, stating that the animals were supplemented for 24 days is irrelevant because your latest measurement of efficacy of the treatment was measured only until day 16. Please clarify and correct as needed.

Line 134: Important to state how far apart were trials 1 and 2, and what measurements were taken to avoid potential carryover effects.

Line 138: what was a new targeted daily supplementation? 0.05%, how was this determined?

Line 148: add information about sample collection, handling, and storage until DNA extraction.

Lines 150-154: was this questionnaire validated? Also, was it a qualitative or quantitative assessment? In other words, did it had a scale that owners would use to grade the parameters? How was, for example, “attitude” measured or its change? Please add a reference if available. Alternatively, you could add the questionnaire used as a supplementary material.

Line 172: did you rarefy the sequences to run alpha and beta-diversity? What was the read length used for it?

Line 199: the percentage of sequences from Bacteroidetes seems pretty low, is it possible that the technique used could have primer biases? If so, this needs to be addressed somewhere n this manuscript (e.g., discussion).

Lines 215-216: where is these data? It is not presented in this manuscript.

Line 219: there was no mention of filtering steps under “Sequence analysis”. This information needs to be described under materials and methods. Was 4,600 the length used in the rarefaction step?

Lines 234-236: Figure 6, LEfSe analysis only showing individual variation, not treatment effect, correct? Is this relevant?

Lines 278-319: the first 2 pages of the discussion section describes individual differences among animals and inconsistent responses even within animal. While this is an important point to be acknowledged a detail explanation of individual responses is not meaningful. Thus, it is recommended for this portion to be shortened highlighted only the main points.

Lines 231-326: if low doses have been shown to be ineffective, why did you select a low dose and not a higher one based on the literature available? A justification of the reason for selecting the doses for trial 1 and 2 is warranted.

Line 309: mention of probiotic literature did not add to the discussion. It was a bit distracting and also seems like a typo.

Line 329: add “dietary” in front of fiber.

Lines 335-337: “massive sequencing”, do you mean high-throughput? It does not allow investigation of all microbial groups, but the majority would be correct. Please revise sentence.

Line 340: “veterinary small animal medicine” are you referring exclusively to cats and dogs or other animal species within the “small animal” group too? Please clarify and revise as needed.

Lines 342-351: what is most critical to provide a constant supplementation based on the BW of the animal or a constant supplementation based on daily food intake? Since you used client-owned animals, it is also possible that animals of same BW would eat very different amounts of food. From your discussion, it is not clear what is your position in minimizing issues with prebiotic supplementation using a scenario of higher variation among animals. Please clarify your point.

Lines 352-354: Please add why/ how is this relevant to others?

Lines 388-390: these changes were not all related to prebiotic supplementation, but largely driven by individual responses, correct? As such, this sentence need to be revised accordingly.

Figure 1: last collection point for fecal sample collection shows as day 16, if this is the case, the supplementation for 24 days is irrelevant and the manuscript should be revised to reflect a 16-day supplementation with the prebiotic source.

Figures 2 and 8: for the boxplots, please add p-values and some symbol that denotes statistical significance when it is applicable.

Figure 4: I think you meant, “dashed lines” instead of “dotted lines”. Also, was the LefSe analysis done using default parameters? It seems different than the default figures generated and scale used. Same comment for figures 6 and 10.

Figures 5 and 7: should p-value be (p>0.05)?

---

## Round 0.2 · accepted · Accept

· Academic Editor

Accept

The authors have done a good job of addressing the comments of the reviewers. The language is much clearer, and the focus in inter-individual variability is appropriate.

Reviewer 2 ·

Basic reporting

The authors did a good job revising and addressing the comments. No further modifications are needed.

Experimental design

No comment

Validity of the findings

No comment

Additional comments

The authors did a good job revising and addressing the comments. No further modifications are needed.